# The Molecular Evolution of Type 2 Vaccine-Derived Polioviruses in Individuals with Primary Immunodeficiency Diseases

**DOI:** 10.3390/v13071407

**Published:** 2021-07-20

**Authors:** Kouichi Kitamura, Hiroyuki Shimizu

**Affiliations:** Department of Virology II, National Institute of Infectious Diseases, 4-7-1 Gakuen, Musashimurayama-shi, Tokyo 208-0011, Japan; kkita@nih.go.jp

**Keywords:** poliovirus, oral poliovirus vaccine, vaccine-derived poliovirus, immunodeficiency-associated vaccine-derived poliovirus, global polio eradication

## Abstract

The oral poliovirus vaccine (OPV), which prevents person-to-person transmission of poliovirus by inducing robust intestinal immunity, has been a crucial tool for global polio eradication. However, polio outbreaks, mainly caused by type 2 circulating vaccine-derived poliovirus (cVDPV2), are increasing worldwide. Meanwhile, immunodeficiency-associated vaccine-derived poliovirus (iVDPV) is considered another risk factor during the final stage of global polio eradication. Patients with primary immunodeficiency diseases are associated with higher risks for long-term iVDPV infections. Although a limited number of chronic iVDPV excretors were reported, the recent identification of a chronic type 2 iVDPV (iVDPV2) excretor in the Philippines highlights the potential risk of inapparent iVDPV infection for expanding cVDPV outbreaks. Further research on the genetic characterizations and molecular evolution of iVDPV2, based on comprehensive iVDPV surveillance, will be critical for elucidating the remaining risk of iVDPV2 during the post-OPV era.

## 1. Introduction

The Global Polio Eradication Initiative has completely eliminated 2 of 3 serotypes of wild polioviruses worldwide with extensive immunization with poliovirus vaccines, the Sabin oral poliovirus vaccine (OPV), and Salk inactivated poliovirus vaccine (IPV) [1]. After the identification of the last polio case caused by type 1 wild poliovirus in Africa in 2016 [2], Pakistan and Afghanistan were the only countries with endemic wild poliovirus since 2017 [3]. The total number of wild polio cases was 140 in 2020 [4].

Trivalent OPV (tOPV), which contains three live-attenuated poliovirus serotypes (usually Sabin 1, 2, and 3 strains), was a crucial tool for global polio eradication by inducing a high level of intestinal immunity to prevent person-to-person transmission of poliovirus in communities. However, due to their intrinsic genetic instability, the OPV strains can evolve into more neurovirulent revertants in the vaccinated individuals and then transmit in communities, which are occasionally associated with polio outbreaks known as circulating vaccine-derived polioviruses (VDPV; cVDPVs) [3,5,6,7,8,9]. All three OPV serotypes can be associated with paralytic polio outbreaks. However, type 2 is the major causative agent of cVDPV outbreaks, producing the highest numbers of events and cases with acute flaccid paralysis (AFP). The number of polio outbreaks caused by type 2 cVDPV (cVDPV2) is on the rise in different geographical areas; thus, suboptimal intestinal immunity against type 2 poliovirus poses a serious public health threat during the final stage of global polio eradication [3,10,11].

Even after cVDPV2 outbreaks were controlled, VDPV infections in immunocompromised individuals, known as immunodeficiency-associated VDPV (iVDPV), remain a potential risk factor for VDPV outbreaks [9,12,13]. The molecular evolution of RNA viruses in chronic infections can lead to the emergence of unique genetic variants with distinct phenotypes from the parental viruses, even for those that are commonly associated with acute viral infections [14,15,16,17]. Thus, chronic infections of RNA viruses are a potential source for future epidemics. In this study, we will examine the prevalence of iVDPV and molecular analysis of iVDPV isolates, especially for type 2 iVDPV (iVDPV2). We then discuss the future challenges in elucidating the molecular evolution and characterization of iVDPV2 and the remaining risk of iVDPV2 in the post-OPV era.

## 2. Definition and Classification of Vaccine-Derived Polioviruses

VDPVs are diverse OPV-derived variants with more than 1% nucleotide divergence in the capsid VP1 region from the corresponding type 1 and type 3 Sabin strains and more than 0.6% nucleotide divergence from the Sabin 2 strain [12]. VDPVs are categorized by the source of samples and epidemiological information, including cVDPVs identified in person-to-person transmission in the community, iVDPVs isolated from persons with primary immunodeficiency (PID), and ambiguous VDPVs (aVDPVs) indicating neither cVDPV nor iVDPV [12,18,19,20]. Some of the PID individuals may excrete iVDPV for a prolonged period. In this study, we classified iVDPV excretors as long-term (6 months to 5 years) or chronic (more than 5 years) [12].

## 3. Status of Polio Outbreaks Due to cVDPV2

As previously mentioned, type 2 OPV strain (OPV2) was mainly associated with cVDPV outbreaks. Recently, the World Health Organization (WHO) implemented a revised polio immunization policy in the Global Polio Eradication Initiative Strategic Plan 2013–2018 [21] that eliminates the type 2 component from tOPV in April 2016 [22], thereby switching from the tOPV to the bivalent OPV (bOPV) including only types 1 and 3. At the same time, WHO encouraged at least one dose of IPV for routine immunization [22,23] to maintain the population immunity against type 2 poliovirus. In addition, a global stockpile of monovalent type 2 OPV (mOPV2) was prepared and maintained for response to cVDPV2 outbreaks.

As expected, immediately after the switch to bOPV, few cVDPV2 epidemics were identified, only in the Democratic Republic of the Congo and Syria, in 2017. However, the number of cVDPV2 outbreaks has been growing since 2018. There were 71 AFP cases in 5 countries in 2018, 366 cases in 16 countries in 2019, and 1054 cases in 24 countries in 2020 (a WHO weekly report as of 1 June 2021) [24]. Molecular epidemiological analysis of cVDPV2 isolates revealed that most of the cVDPV2 outbreaks were derived from supplemental immunization with mOPV2 after switching to bOPV in 2016, presumably in areas with suboptimal intestinal immunity against type 2 poliovirus [8,25].

There is a critical dilemma in the creation of new cVDPV2 outbreaks by mOPV2 immunization in high-risk areas. Novel type 2 OPVs (nOPV2s), type 2 OPV candidates more genetically stable than the conventional Sabin 2 strain, were developed and introduced in some countries with endemic cVDPV2 transmissions as emergency responses to the ongoing cVDPV2 outbreaks [26,27,28]. The coronavirus disease 2019 (COVID-19) pandemic disrupted the surveillance and immunization activities for vaccine-preventable diseases, including polio, since 2020 [11,29]. However, nOPV2 was already introduced to two cVDPV2-affected countries, Nigeria and Liberia in Africa, by the end of May 2021 [30]. The risk of transmission of type 2 poliovirus will be significantly reduced after controlling the cVDPV2 outbreaks. However, the presence of chronic iVDPV2 excretors will remain a potential source of cVDPV2 transmission in communities with lowered intestinal immunity against type 2 poliovirus.

## 4. Prevalence of iVDPV2-Positive Cases

According to Macklin et al. [12], 149 cases of iVDPV (with or without AFP) were reported to the WHO from 1961 to 2019. About 60% of the iVDPV cases involved the type 2 OPV strain, including 83 (56%) single iVDPV2 infections, 3 (2%) infections with a mixed serotype of types 1 and 2, and 3 (2%) infections with a mixed serotype of types 2 and 3. After the switch from tOPV to bOPV in 2016, the risk of iVDPV2 emergence was expected to decrease substantially. In fact, only two iVDPV2 cases were identified during 2017–2019 [12,31], including a chronic iVDPV2 excretor in the Philippines (see Section 6).

Among the different types of PID, patients with common variable immunodeficiency (CVID) or severe combined immunodeficiency (SCID) are associated with higher risks for long-term iVDPV infections; thus, they are less likely to clear iVDPVs spontaneously before death. However, only a limited number of chronic iVDPV excretors were reported, partly due to the low probability of survival of the PID patients and insufficient surveillance to monitor the iVDPV cases with or without AFP, especially in low-income countries. Among the 10 reported cases of chronic iVDPV excretors (Table 1), 8 were reported to be CVID, and 5 were associated with iVDPV2 [12,32,33]. Among the patients with predominantly antibody deficiencies, CVID patients are more associated with asymptomatic and prolonged iVDPV infection than those with agammaglobulinemia and hypogammaglobulinemia [32].

## 5. Genetic Analysis and Molecular Characterization of iVDPV2 Isolates

### 5.1. Available Sequence Dataset of iVDPV2 Isolates

The sequence information on cVDPV isolates is expanding due to the current cVDPV outbreaks [34]. Accordingly, there is increasing information on the molecular evolution of cVDPV [18,25,35]. However, it remains difficult to characterize iVDPVs genetically and elucidate the trends in the molecular evolution of iVDPV because of the limited availability of the sequences and clinical information of iVDPV and the lack of appropriate annotation of each iVDPV case [36,37]. The histories of the patients with long-term iVDPV infections vary greatly in terms of the duration of infection, immunological status, treatment and formulation of intravenous immunoglobulin products, and the kinds of clinical specimens. There are also lingering concerns about the reliability of molecular analyses using limited and potentially biased sequence datasets of the iVDPV isolates. Moreover, very little information on the capsid or full-genome sequences of the iVDPVs is available for investigating the evolutionary dynamics of iVDPV in the microenvironment in a long-term iVDPV excretor.

Therefore, in this study, we have updated the iVDPV2 sequence dataset in reference to the previous reports [36,37,38,39,40,41,42,43,44,45,46,47] (Table 2). Our dataset is based on the sequence information in GenBank; thus, some of the clinical and epidemiological information is incomplete. In addition, it should be noted that there is no direct evidence of long-term excretion in the suspected iVDPV isolates from the environmental samples, although the involvement of iVDPV excretion has been highly suggested [43,44]. According to the phylogenetic analysis of the VP1 regions in our iVDPV2 dataset, the iVDPV2 isolates derived from a certain long-term iVDPV excretor are genetically closely related to each other and form a unique genetic cluster, and are not related to those from other iVDPV cases (Figure 1). In addition, these iVDPV2 sequences are not genetically related to those of the cVDPV2 or aVDPV2 available in the GenBank database (data not shown).

### 5.2. Molecular Evolution of iVDPV2

During the community transmission of type 1 wild polioviruses, the rate of nucleotide substitution at all the sites in the entire capsid P1 region was estimated to be 1.03 × 10^−2^ substitutions/site/year [49]. Similarly, the average estimated substitution rate was 1.14 × 10^−2^ substitutions/site/year for the major genetic lineages of cVDPV2 isolates in Nigeria [18]. In the case of iVDPV2, the rate of nucleotide substitution in the VP1 region was 1.51 × 10^−2^ substitutions/site/year for the most long-term iVDPV2 excretor in the United Kingdom (for approximately 28 years at the time) [45]. Shaghaghi et al. observed rapid molecular evolution in some iVDPV cases at the initial stages of virus replication after OPV administration [32]. After the initial replication period of the OPV strains, the subsequent mutations would accumulate at a nearly uniform rate of 1 to 2% nucleotide changes per year [32,50,51]. Due to a strong positive selection in vaccinees immunized with type 2 OPV, the so-called gatekeeper mutations from the attenuated Sabin 2 strain, including three mutations at A481G, U2909C (VP1-I143T), and U398C, are rapidly selected and fixed in the first several weeks post-vaccination [35,52,53]. Those gatekeeper mutations may contribute to the virus replication fitness in the human intestine and higher initial evolution rates of iVDPV and cVDPV immediately after the OPV administration.

### 5.3. Amino Acid Substitutions in Phenotypic Determinants

Consistent with Zhao et al. [37], it is difficult to distinguish between the iVDPV and cVDPV strains in our iVDPV2 dataset by only comparing the nucleotide and amino acid sequences in the VP1 region (data not shown). However, a codon-by-codon comparison throughout the VP1 region using SNAP [54] showed that the rates of non-synonymous substitutions in the iVDPV strains were higher than those in cVDPV2 (Figure 2). These data suggest that the molecular evolution of the capsid proteins associated with phenotypic determinants, including antigenic sites, is more likely to occur in the iVDPV strains than in the cVDPV strains, especially for long-term iVDPV infections. Among the 19 iVDPV2 cases, amino acid substitutions are frequently identified in some of the neutralizing antigenic sites (NAgs) and a hypervariable region at the N-terminus of VP1 and VP1–143 in the non-NAg sites (Figure 3). The N-terminus of VP1 forms an amphipathic helix and is presumably involved in the interaction with the host cell membrane. Regardless of the high variability of this domain, the predicted amphipathic helix is maintained [55]. McDonald et al. reported a decline in neutralizing capacity of human sera immunized with IPV and OPV against the highly evolved iVDPV2 from long-term excretors due to the amino acid substitutions in the NAgs [36]. On the other hand, there was no significant change in the neutralization titers of the human sera against a series of iVDPV2 variants from a chronic iVDPV2 excretor in the United Kingdom, compared with those against the parental Sabin 2 strain, regardless of amino acid substitutions in the NAgs in the Sabin 2 strain [46]. No significant changes in neutralizing capacity were observed in the highly evolved cVDPV2 isolates in Nigeria compared to those in the Sabin 2 strain, suggesting that the changes in the NAgs of the cVDPV2 were lower than those in the iVDPV2 [36,37,55] (Figure 4). Non-synonymous substitutions frequently occurred not only in the NAg sites but also in the non-NAg sites located on the surface of the capsid (Figure 4), potentially affecting the antigenicity or in vivo fitness of the VDPVs.

G481 and VP1-I143T were identified as the major attenuation determinants for the Sabin 2 strain [9,19]. As previously mentioned, these two sites are gatekeeper mutations that are rapidly selected and fixed immediately after the administration of OPV2 [35]; most of the iVDPV2 isolates very likely have those mutations in the attenuation sites. In fact, a series of highly evolved iVDPV2 isolates from a chronic excretor in the United Kingdom are more neurovirulent than the parental Sabin 2 strain in a human poliovirus receptor transgenic mouse model [45].

### 5.4. Genomic Recombination

Most of the cVDPV isolates associated with polio outbreaks are recombinants of OPV-derived capsid sequences and non-capsid sequences derived from non-polio enteroviruses (NPEVs), particularly those belonging to *Enterovirus C* [9,18,19,20,56,57,58,59,60,61]. To our knowledge, there is no concrete evidence of long-term excretion or local transmission of iVDPV strains recombinant with NPEV in part due to limited information on the full-genome sequences of iVDPV isolates. The recombinant viruses derived from the Sabin 2 capsid sequences with the non-capsid sequences from Sabin 3 or Sabin 1 are frequently detected in healthy vaccinees and VAPP cases in the early stages of virus replication after tOPV immunization [55,62,63]. The recombination between different genetic lineages among genetically divergent iVDPV variants was identified in a long-term type 1 iVDPV excretor in Taiwan [64]; therefore, recombinations, as well as mutations, may contribute to the molecular evolution and viral fitness in the microenvironment of iVDPV infection. However, the virological significance of viral recombination in the host is still uncertain because only a few reports based on the full-genome sequences of iVDPV are available.

### 5.5. In Vivo Fitness and Quasi-Species

In individuals with long-term iVDPV infections, genetically diverse iVDPV variants may co-exist as viral quasi-species. The role of quasi-species in the molecular evolution of iVDPV remains unclear; however, in general, the genetic diversity of polioviruses within infected individuals is known to be involved in the in vivo molecular evolution and the rise of certain viral phenotypes, including neurovirulence [65,66,67]. Historically, poliovirus quasi-species were analyzed by the conventional plaque cloning and Sanger sequencing of virus isolates. Recently, next-generation sequencing (NGS) technologies were applied to study viral quasi-species and the molecular evolution of various RNA viruses, including those of the OPV strains and cVDPVs, but not for those from iVDPV cases [25,52,53]. In the future, more comprehensive and quantitative analyses on the genetic diversity of the quasi-species of iVDPV variants in infected individuals, compared to those in cVDPV and OPV strains, using NGS will be required.

## 6. Current and Future Risk of iVDPV2

Although there is no substantial evidence of polio outbreaks caused by cVDPV2 derived from iVDPV2 excretors, the relative risk of long-term iVDPV2 infection will rise as the population with lower intestinal immunity to type 2 poliovirus among the individuals immunized with bOPV and IPV increase [10,13,68,69]. The WHO Western Pacific Region (WPR) was certified to be wild polio-free since 2000, except for several importations of wild polioviruses from endemic countries [70,71,72]. Since 2001, several cVDPV outbreaks were reported in the WPR; however, all of them were effectively controlled using supplemental OPV immunization [60,73,74]. In September 2019, type 1 VDPV isolates (more than 3% nucleotide divergence from the Sabin 1 strain in the VP1 region) were identified from environmental sewage samples collected in the National Capital Region (NCR) in the Philippines. Highly evolved type 2 VDPV isolates (more than 7% nucleotide divergence from the Sabin 2 strain in the VP1 region) were concomitantly detected in some of the same sewage samples [8,25].

Meanwhile, more extensive AFP and environmental surveillance identified many VDPV1 and VDPV2 isolates from AFP cases and environmental samples from different geographical areas, confirming that the simultaneous and widespread transmission of cVDPV1 and cVDPV2 was associated with paralytic polio outbreaks in the Philippines and Malaysia in 2019–2020. At the same time, intensified surveillance identified highly divergent VDPV2 isolates (more than 7% nucleotide difference in the VP1) from a 5-year-old male with reduced antibody levels who had received three doses of tOPV from August 2014 to February 2015 in Laguna province, close to the NCR in the Philippines [12]. Follow-up samplings demonstrated that genetically related VDPV2 variants were continuously detected in stool samples of the patient from August 2019 to now (as of May 2021), indicating that the patient is a chronic iVDPV2 excretor.

At least in the VP1 region, the cVDPV2 isolates in the Philippines in 2019–2020 are not genetically related to the iVDPV2 isolates from the chronic excretor in Laguna. Macklin et al. estimated that the cVDPV2 isolates in the Philippines were seeded by tOPV immunization around 2014 before the switch to bOPV in 2016 [25]. According to the genetic diversity of the iVDPV2 isolates from the Sabin 2 strain and the history of OPV immunization of the patient, the chronic iVDPV2 infection might have been initiated in 2014–2015 at almost the same time with the emergence of cVDPV2. The origin and evolution of both cVDPV2 and iVDPV2 remain uncertain epidemiologically and genetically, partly due to the lack of any iVDPV2 and cVDPV2 cases for nearly 5 years in the Philippines.

Recently, Valesano et al. found that most OPV2-derived variants with the gatekeeper mutations that appeared in OPV recipients rapidly could not survive during virus transmission to the close contacts of the recipients due to the tight transmission bottleneck. This finding suggests distinct mechanisms of molecular evolution between viral replication in the hosts and during community transmission [52]. More detailed and comprehensive genetic characterization, taking the previously mentioned points on the molecular characteristics of iVDPV2 into account, is in progress to elucidate the relationship and molecular evolution of cVDPV2 and iVDPV2 in the Philippines.

## 7. Conclusions

The last cVDPV2 isolate in the Philippines was detected from an environmental sample collected in January 2020. No cVDPV2 was identified from AFP and environmental samples after that. However, the incidences of cVDPV2 and iVDPV2 in the Philippines and Malaysia in 2019–2020 highlight the risk of inapparent infections or silent VDPV transmission or both, even in areas with a long-standing polio-free status. Further research on the genetic characterization and molecular evolution of iVDPV2 will enable us to mitigate the remaining risk of widespread transmission of iVDPV2 during the post-OPV era.

## Figures and Tables

**Figure 1 viruses-13-01407-f001:**
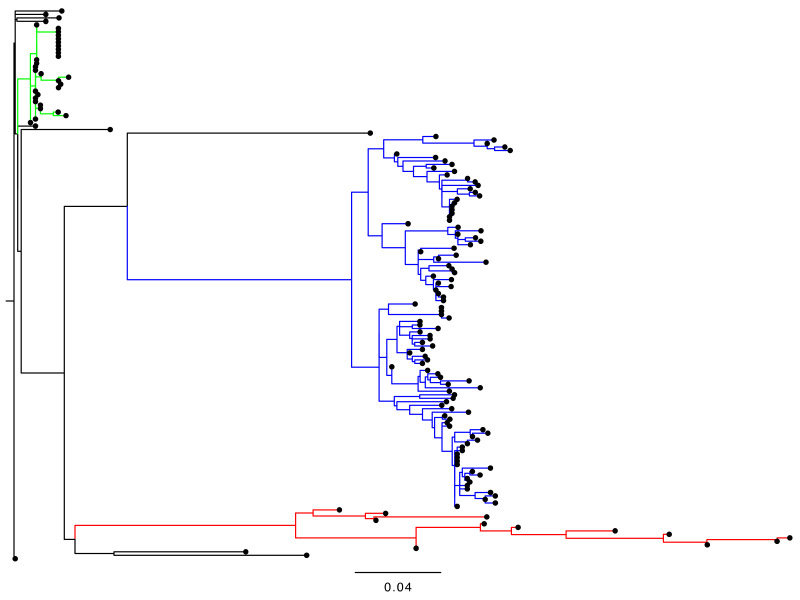
The phylogenetic tree of VP1 sequences in 157 iVDPV2 isolates. The sequence alignment was constructed using Clustal Omega, and the tree was inferred using the FastTree 2.1.10 [48] with the GTR model and gamma distribution and visualized with FigTree. Large genetic clusters of each case study are indicated by different colors, red for a chronic excretor in the United Kingdom 1995–2013 [45], blue for the environmental samples from Slovakia in 2003–2005 [43], and green for the stool and oropharyngeal samples from Israel in 2018 [46,47].

**Figure 2 viruses-13-01407-f002:**
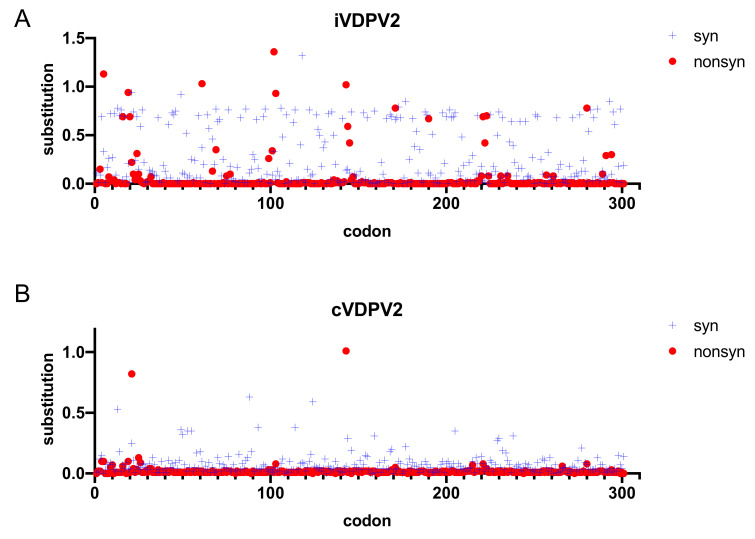
The frequency of synonymous and non-synonymous substitutions of the VP1 amino acid sequences from 157 iVDPV2 and 567 cVDPV2 isolates. Sequence divergence was analyzed using SNAP (Synonymous Non-synonymous Analysis Program, https://www.hiv.lanl.gov/content/sequence/SNAP/SNAP.html (accessed on 28 May 2021)) [54]. The synonymous (blue crosses) and non-synonymous substitutions (red circles) of the VDPV sequence relative to the Sabin 2 strain are depicted. (**A**) iVDPV2. (**B**) cVDPV2.

**Figure 3 viruses-13-01407-f003:**
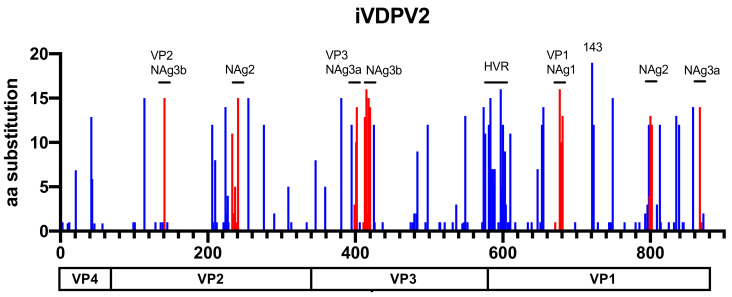
The number of amino acid substitutions in each position of the P1 capsid region of 19 iVDPV2 isolates. The amino acid substitutions in the neutralizing antigenic (NAg) sites are shown in red. The positions of hypervariable residues (HVRs) at the N-terminal and a gatekeeper substitution at VP1–143 of VP1 are indicated.

**Figure 4 viruses-13-01407-f004:**
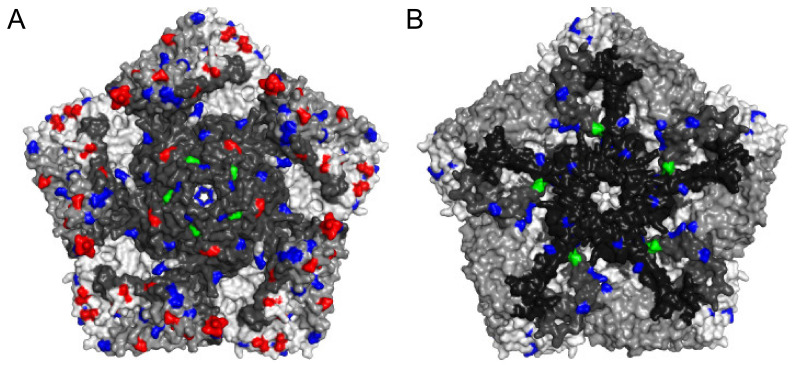
The locations of amino acid substitutions in selected iVDPV2 isolates. The three-dimensional structure of type 2 wild poliovirus strain MEF-1 was obtained from PDB 1EAH and represented as a pentameric unit ((**A**): outside; (**B**): inside of capsid). Each protomer contains a single copy of VP1, VP2, VP3, and VP4 distinguished by gray colors. The location of frequent amino acid substitutions (more than 4 out of 19 iVDPV2 isolates) relative to the Sabin 2 strain is colored. The neutralizing antigenic sites are red. The hypervariable residues and the gatekeeper substitution of VP1-143 are green. Other amino acid changes are blue. The image was generated using PyMOL 2.4.2 (Schrödinger).

**Table 1 viruses-13-01407-t001:** Chronic iVDPV excretors.

Case No.	Year Detected	Localization	PID Type	Serotype	Maximum VP1 Divergence (%)	Estimated Replication Period (years)	Reference
1	1981	USA	CVID	1	10 ^†^	7.6 ^†^	[32]
2 ^§^	1986 1992	USA	CVID	1 2	5.4 ^†^ 11.8 ^†^	4.7 ^‡^ 9.6 ^†^	[32,33]
3	1990	Germany	CVID	1	8.3 ^†^	9.5 ^†^	[32]
4	1995	UK	CVID	2	17.9 ^†^	27.83 ^†^	[32]
5	2000	Germany	CVID	1	8.5 ^†^	8.5 ^†^	[32]
6	2002	UK	CVID	2	6.3 ^†^	6.3 ^†^	[32]
7	2009	India	CVID	1	5.2 ^†^	5 ^†^	[32]
8	2009	USA	CVID	2	12.3 ^†^	11.9 ^†^	[32]
9	2015	India	SCID	3	10.2 ^†^	6 ^†^	[32]
10	2019	Philippines	Hypokalemia and infectious diarrhea	2	7.6 ^||^	5.0 ^||^	[12]

^†^ Referenced from Supplementary Table 4 of Reference [32]. ^‡^ Referenced from Table 2 of Reference [33]. ^§^ Two iVDPVs were isolated from the same patient at different times. ^||^ Referenced from Table 2 of Reference [12]. CVID: common variable immunodeficiency. SCID: severe combined immunodeficiency.

**Table 2 viruses-13-01407-t002:** Sequence dataset of probable type 2 iVDPV strains.

GenBank Accession No.	Genome Region	Year Detected	Localization	Source of Samples	VP1 Divergence (%)	Estimated Replication Period (years)	Reference
GU390704	P1	1992	USA	NA	10.4	NA	[39]
KR817050-817060 AJ544513	P1 Full genome	1995-2013	UK	Chronic excretor (Case 4 in Table 1)	17.9	28	[45]
AJ288062	P1	1998	Israel	Environmental samples	9.4	NA	[44]
AY177685	Full genome	2000	Italy	iVDPV2 case	0.88	1.42	[42]
DQ890387	Full genome	2002	Nigeria	iVDPV2 case	2.5	1.5	[41]
JX913541-913647	VP1	2003-2005	Slovakia	Environmental samples	3.4	NA	[43]
FJ517648	P1	2007	Belarus	iVDPV2 case (AFP)	1.88	1.6	[40]
FJ517649	P1	2007	Russia	iVDPV2 case (AFP)	1.44	NA (no OPV history)	[40]
GU390707	Full genome	2009	USA	Chronic excretor (Case 8 in Table 1)	11.83	11.9	[39]
KR709241	VP1	2013	Germany	iVDPV2 case	1.0	0.5–0.9	[38]
KR709242	VP1	2013	Germany	iVDPV2 case	4.4	2.4–2.8	[38]
MK660464-660492	VP1	2015	Israel	iVDPV2 case	>0.8	NA	[46,47]

NA: not available.

## Data Availability

All data of this study are available within this manuscript (Table 2).

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
