# Peer review of "The Molecular Evolution of Type 2 Vaccine-Derived Polioviruses in Individuals with Primary Immunodeficiency Diseases"

_viruses, 2021, doi:10.3390/v13071407_

Round 1

Reviewer 1 Report

The benefits of polio vaccination campaigns aimed at eliminating the disease and the pathogen poliovirus are threatened by the development of polio cases due to pathogenic strains derived from the oral polio vaccine (VDPVs). These strains originate from and circulate in poorly vaccinated populations (cVDPVs). VDPVs also found in immunodeficient patients (mainly antibody deficiency) who may be infected long term or chronically (iVDPVs). Unlike cVDPVs, these iVDPVs are not yet involved in human-to-human transmission but could possibly maintain a reservoir of pathogenic poliovirus strains.

The article by K. Kitamura and H. Shimizu was based on the exhaustive data available about type 2 iVDPVs (the most common iVDPVs) and in particular, on sequencing data. This study reviewed the iVDPVs characteristics, highlighting those that could differentiate iVDPVs from cVDPVs. Among essential results, phylogenetic analyzes of the sequences showed that the different iVDPVs variants gather in clusters that clearly differ from one patient to another. In addition these clusters are not related to those of cVDPVs. Furthermore, the genetic diversity of iVDPVs affected peptide sequences much more often than that of cVDPVs. In particular, the residues participating in the constitution of antigenic neutralization sites were frequently modified.

 All of the results presented in this article constitute a very important assessment for ongoing studies aimed at determining whether iVDPVs can be the cause of polio epidemics (human-to-human transmission) or not. These data are indeed essential to perfect the surveillance of the disease and its pathogenic agent.

The article is clear and well written and can be understood and appreciated by a large audience. Only a few minor comments deserve to be taken into account.

Minor comments:

  • Lines 138-140;  : This sentence doesn’t seem so clear.

  • Lines183-186: What is the biological role of the N-terminus of VP1 that appears to be so variable?

  • Lines 224-227: This sentence deserves to be reworded (…or both?).

  • Lines 234-236: This referee did not understand the meaning of this sentence.

Author Response

The benefits of polio vaccination campaigns aimed at eliminating the disease and the pathogen poliovirus are threatened by the development of polio cases due to pathogenic strains derived from the oral polio vaccine (VDPVs). These strains originate from and circulate in poorly vaccinated populations (cVDPVs). VDPVs also found in immunodeficient patients (mainly antibody deficiency) who may be infected long term or chronically (iVDPVs). Unlike cVDPVs, these iVDPVs are not yet involved in human-to-human transmission but could possibly maintain a reservoir of pathogenic poliovirus strains.

The article by K. Kitamura and H. Shimizu was based on the exhaustive data available about type 2 iVDPVs (the most common iVDPVs) and in particular, on sequencing data. This study reviewed the iVDPVs characteristics, highlighting those that could differentiate iVDPVs from cVDPVs. Among essential results, phylogenetic analyzes of the sequences showed that the different iVDPVs variants gather in clusters that clearly differ from one patient to another. In addition these clusters are not related to those of cVDPVs. Furthermore, the genetic diversity of iVDPVs affected peptide sequences much more often than that of cVDPVs. In particular, the residues participating in the constitution of antigenic neutralization sites were frequently modified.

 All of the results presented in this article constitute a very important assessment for ongoing studies aimed at determining whether iVDPVs can be the cause of polio epidemics (human-to-human transmission) or not. These data are indeed essential to perfect the surveillance of the disease and its pathogenic agent.

Responses to reviewer’s comments

Thank you for the comments to the previous version of manuscript. According to the comments and suggestions, we have revised the manuscript and the revised words/sentences are indicated in red in the text attached (The line numbers of the revised Word file is not the same with the previous ones). 

The article is clear and well written and can be understood and appreciated by a large audience. Only a few minor comments deserve to be taken into account.

Minor comments:

  • Lines 138-140;  : This sentence doesn’t seem so clear.

response:

We revised the sentence as follow.

In addition, any cVDPV2 or aVDPV2 sequences that were not genetically related to those of the iVDPV2 sequences were identified in the GenBank database (data not shown). →

In addition, these iVDPV2 sequences are not genetically related to those of the cVDPV2 or aVDPV2 available in the GenBank database (data not shown).

  • Lines183-186: What is the biological role of the N-terminus of VP1 that appears to be so variable? 

    response:

    We added the following sentences.

    The N-terminus of VP1 forms amphipathic helix and presumably involved in the interaction with the host cell membrane. Regardless of the high variability of this domain, the predicted amphipathic helix is maintained [55].

  • Lines 224-227: This sentence deserves to be reworded (…or both?). response:

    We revised the sentences as follow.

    Although the full-genome sequences of iVDPV isolates remain highly limited, to our knowledge, there is no concrete evidence of long-term excretion or local transmission of iVDPV strains recombinant with NPEV or both. →

    To our knowledge, there is no concrete evidence of long-term excretion or local transmission of iVDPV strains recombinant with NPEV in part due to limited information on the full-genome sequences of iVDPV isolates.

  • Lines 234-236: This referee did not understand the meaning of this sentence.

    response:We revised the sentences as follow.

    However, the virological significance of recombination in infected individuals is still uncertain because information on the full-genome sequences of iVDPV variants from long-term excretors remains highly inadequate. → 

    However, the virological significance of viral recombination in the host is still uncertain because only a few reports based on the full-genome sequences of iVDPV are available.

Reviewer 2 Report

The paper presents a review of data from Immunodeficient associated vaccine derived poliovirus (iVDPV). Although the number of cases, number of virus sequences and number of environmental samples associated with  iVDPV is very small, the authors do make a very good job at systematically and comprehensively present the available data in a clear way. The authors also make a good analysis of the available sequences leading to interesting new insights into the mutation pressures under way.

The paper is well written and no major flaws were detected. I therefore recommend the publication as is.

Author Response

The paper presents a review of data from Immunodeficient associated vaccine derived poliovirus (iVDPV). Although the number of cases, number of virus sequences and number of environmental samples associated with  iVDPV is very small, the authors do make a very good job at systematically and comprehensively present the available data in a clear way. The authors also make a good analysis of the available sequences leading to interesting new insights into the mutation pressures under way.

The paper is well written and no major flaws were detected. I therefore recommend the publication as is.

Responses to reviewer’s comments

Thank you for the comments to the previous version of manuscript. According to the comments and suggestions, we have revised the manuscript and the revised words/sentences are indicated in red in the text attached (The line numbers of the revised Word file is not the same with the previous ones).